# Taxonomy, Bio-Ecology and Insecticide Resistance of Anopheline Vectors of Malaria in Sri Lanka

**DOI:** 10.3390/ijerph21070814

**Published:** 2024-06-21

**Authors:** Sinnathamby N. Surendran, S. H. P. Parakrama Karunaratne

**Affiliations:** 1Department of Zoology, University of Jaffna, Jaffna 40000, Sri Lanka; noble@univ.jfn.ac.lk; 2Department of Zoology, University of Peradeniya, Peradeniya 20400, Sri Lanka

**Keywords:** anopheline mosquitoes, bio-ecology, insecticide resistance, malaria, sibling species, species complex, vectors, Sri Lanka

## Abstract

The objective of this review was to update the current knowledge on major malaria vectors in Sri Lanka and their bio-ecology and insecticide resistance status. Relevant data were collected through a comprehensive literature search performed using databases such as PubMed, NIH, Google Scholar and Web of Science. Sri Lanka had been endemic to malaria for centuries. However, due to a coordinated public health effort last indigenous malaria case was reported in 2012 and the island nation was declared free of malaria in 2016. Although 25 anopheline mosquitoes have been reported so far on the island, only *Anopheles culicifacies* and *An. subpictus* have been established as primary and secondary vectors of malaria respectively. Both vector species exist as a species complex, and the sibling species of each complex differ in their bio-ecology and susceptibility to malaria parasites and insecticides. The article provides a comprehensive and updated account of the bio-ecology and insecticide resistance of malaria vectors and highlights the challenges ahead of retaining a malaria-free status.

## 1. Introduction

Sri Lanka is an island nation with an area of 65,525 km^2^ located in close proximity to South India, separated by a narrow streak of the Palk Strait (Figure 1). The island is divided into four climatic zones, namely semi-arid, dry, intermediate and wet, based on annual rainfall, being received mainly from two monsoons called Southwest (April to June) and Northeast (October to January) [1]. The wet zone receives an average rainfall of >250 cm, mainly from the Southwest monsoon, whilst the dry zone receives 125–175 cm mainly from the Northeast monsoon. The intermediate zone lies between the wet and dry zones with mixed characteristics and receives 175–250 cm and the semi-arid zone receives an annual rainfall between 63–125 cm [1,2,3].

The nation has experienced severe epidemics due to malaria and has been considered as malaria endemic until recently. Dry and intermediate zones were typically endemic for malaria. There is historical evidence to show the devastating effects of malaria since the 13th century [1,2]. However, there are records of periodic epidemics of malaria during the 20th century [1,2,3,4]. The major malaria epidemic occurred during the 1934–1935 period infecting approximately 5 million people with nearly 85,000 deaths [2]. As a malaria control effort, the Antimalaria Centre was established in Kurunagala in 1911 and the implementation of malaria control programmes was initiated. Malaria control activities were mainly confined to rural areas as malaria in Sri Lanka had been mainly associated with rural irrigation and riverine systems [2].

Both *Plasmodium vivax* and *P. falciparum* contributed to the malaria cases with the former as the predominant species. Soon after the introduction of DDT in 1948, a rapid decline in malaria cases was observed, bringing its tally to six indigenous cases and eleven imported cases in 1963. Because of negligence in continuing surveillance, case detection, and DDT spraying, another epidemic occurred in 1967–1968 with over 500,000 cases and five deaths [2,5]. With the resurgence of malaria, the country implemented a variety of control strategies, and in 1998 the Roll Back Malaria Initiative of the World Health Organisation was employed. Malaria cases dropped sharply from 264,549 in 1999 to 198 in 2007 [5]. Sri Lanka entered the pre-elimination phase in 2008 and the elimination phase in 2011. The last indigenous cases were reported in 2012 and the World Health Organisation declared Sri Lanka free of malaria in 2016 [5]. Although Sri Lanka is free of indigenous malaria, there are reports of imported cases in the form of travellers visiting malaria-endemic countries [1,3,5].

## 2. Materials and Methods

This review was conducted via a search strategy to include relevant information related to malaria vectors from Sri Lanka and their bio-ecology and insecticide resistance. Data were collected using various online database flat forms such as PubMed, National Library of Medicine (NIH), Google Scholar and Web of Science. The used key words were “malaria in Sri Lanka”, “malaria vectors of Sri Lanka”, “anopheline species complexes of Sri Lanka”, and “insecticide resistance of malaria vectors of Sri Lanka”. After a comprehensive analysis, information from the most relevant research publications was compiled.

The article is structured to initially provide a brief overview of the anopheline species of Sri Lanka and the major malaria vectors in Sri Lanka. As only two species, namely *Anopheles culicifacies* and *An. Subpictus*, have been implicated as primary and secondary vectors respectively, the major emphasis was given to these two species. With the background that the two major species exist as species complexes, the article gives a comprehensive update on the existence of sibling species and their bio-ecology and insecticide resistance. Relevant information on minor vector species is also given to provide additional knowledge to the readers. The article also highlights recent studies on the molecular characterisation of sibling species and the discrepancy in morphology-based identification. An extensive coverage on the resistance development of malaria vectors to insecticides, commonly used for malaria control programmes, is presented highlighting the underlying mechanisms responsible for resistance. Challenges in maintaining a malaria-free status with the recent invasion of *An. stephensi*, a major concern for malaria control, is discussed towards the end of the review.

## 3. Anopheline Species in Sri Lanka

The taxonomy and identification of Sri Lankan anopheline mosquitoes was initiated in 1925 and in 1950 a list of mosquitoes of Sri Lanka was published. Later, the list was updated in 1981 and 1990 [2]. Twenty-five anopheline species have been recorded so far in Sri Lanka. Nine species, namely *Anopheles aitkenii* James, 1903; *An. barbirostris* Van der Wulp, 1884; *An. barbumbrosus* Strickland & Choudhury, 1927; *An. gigas refutans* Alcock, 1913; *An, interruptus* Purim 1929; *An. nigerrimus* Giles, 1900; *An. peditaeniatus* Leicester, 1908; *An. peytoni* Kulasekera, Harrison & Amerasinghe, 1988; and *An. reidi* Harrison, 1973, have been grouped under the subgenus Anopheles Meigen. Sixteen species, namely *An. aconitus* Donitz, 1902; *An. annularis* Van der Wulp, 1884; *An. culicifacies* Giles, 1901; *An. elegans* James, 1903; *An. jamesii* Theobald, 1903; *An. jeyporiensis* James, 1902; *An. karawari* James, 1902; *An. maculatus* Theobold, 1901; *An. mirans* Sallum & Peyton 2005; *An. pallidus* Theobold, 1901; *An. pseudojamesi* Strickland and Chowdhury, 1927; *An. subpictus* Grassi, 1899; *An. stephensi* Liston, 1901; *An. tessellatus* Theobald, 1901; *An. vagus* Donitz, 1902; and *An. varuna* Iyengar, 1924 have been classified under the subgenus Cellia Theobald [6,7,8,9,10].

## 4. Malaria Vectors

*Anopheles culicifacies* sensu lato (s.l.) was incriminated as a malaria vector for the first time in the country in 1913 [2]. Since then, the species has been well established as the primary vector throughout the island [2,11]. *Anopheles subpictus* sensu lato has been implicated as an important secondary vector in many parts of the country [12]. Malaria parasites have been detected in the field-collected samples of *An. aconitus*, *An. annularis*, *An. nigerimus*, *An. pallidus*, *An. tessellatus*, *An. vagus*, *An. varuna*, *An. barbirostris* and *An. peditaeniatus*, in addition to the primary and secondary vector species, using mosquito dissections and enzyme-linked immunosorbent assays [12,13,14,15].

Malaria vector prevalence in Sri Lanka is associated with climatic conditions and local ecology. Since anopheline mosquitoes undergo preimaginal development in freshwater habitats, anopheline mosquito prevalence is recorded mainly along riverine systems, irrigation canals, lakes and ponds. The development of irrigation projects has also contributed to the distribution and prevalence of anopheline mosquitoes. Irrigation projects associated with the Mahaweli Development Project had an impact on the local malaria epidemic during the 1980s in the Polannaruwa district (Figure 1) by supporting the proliferation of certain anopheline mosquitoes (e.g., *An. annularis*) to function as efficient vectors [1,2].

## 5. Anopheline Species Complexes and Sibling Species Vectors

Morphologically, more or less similar and reproductively isolated populations within a taxon are called sibling species and the taxon is called a species complex [16,17]. Sibling species are also called isomorphic or cryptic species [17,18]. Sibling species of a species complex show different bio-ecological traits that include variation in the distribution, feeding and resting behaviour and susceptibility to parasites and insecticides [19]. There are a number of malaria vector mosquitoes that exist as species complexes across the world. It is important to identify species complex and its members of malaria vectors as the presence of two or more unidentified and/or uncharacterized sibling species of a species complex may influence real malaria transmission patterns, leading to sub-optimal vector control programs [20]. Therefore, it is crucial to identify sibling species and characterise their bio-ecology to launch optimum vector control programmes avoiding the waste of resources by targeting non-malaria vector species in a species complex [19,20].

### 5.1. Anopheles culicifacies Complex in Sri Lanka

*Anopheles culicifacies* s.l. is the primary vector of malaria in Sri Lanka. This major vector species is composed of five sibling species, namely A, B, C, D and E, and is widely prevalent on the Indian subcontinent [19]. The sibling species A–D were initially identified based on the inversion pattern on the polytene chromosome [19]. Sibling species B and E share the same banding pattern in the polytene chromosome but differ in Y-chromosome morphology, in which the former is acrocentric and the latter is sub-metacentic [21]. While sibling species A, C and D are considered potential vectors, sibling species B is regarded as a poor vector [19,20,21].

Based on initial characterization compatible to the Indian polytene chromosome banding pattern, only species B, which is a poor vector of malaria in India, was reported to be present in Sri Lanka [22]. This conundrum was subsequently resolved through the identification of species E in Sri Lanka, based on mitotic Y-chromosome variation [23]. Since sibling species B and E can only be differentiated based on Y-chromosome morphology, identification of field-caught species B and E females depends on the examination of mitotic chromosomes of their F1 progeny which is a cumbersome procedure [20].

Extensive molecular analysis based on ITS2 and COII sequence variations showed that Sri Lankan sibling species B and E have sequence similarity at multiple loci and found that the karyotypic assignment of sibling species B and E failed to correlate with cytochrome oxidase subunit I and microsatellite genotypes [24,25]. However, the genetic structure analysis resulted in two genetic clusters, not associated with karyotypes, indicating the existence of two genetically different populations of the Culicifacies Complex in Sri Lanka that are not associated with the Y-chromosome karyotype [25]. A PCR-based assay is available to distinguish the Indian sibling species B and E [26]. However, the applicability of this assay is to be evaluated for Sri Lanka B and E, identified based on mitotic chromosome morphology. However, the reported genetic variations have not been so far associated with malaria transmission and vector control in Sri Lanka may be due to the fact that no indigenous cases have been reported since 2012.

#### Bio-Ecology of the Sibling Species of Culicifacies Complex in Sri Lanka

The distribution of species B and E, identified based on Y-chromosome morphology in Sri Lanka, is shown in Figure 1 [20]. Although molecular similarities are reported between sibling species B and E in Sri Lanka, there are findings that show both species differ in their bio-ecological traits (Table 1). While both siblings are sympatric, species E dominates over species B in many parts of the country including the dry and intermediate zones (Figure 1) [20]. While both species prefer to feed and rest indoors, there is no clear data to show their relative anthrophagicity and zoophagicity [20].

*Anopheles culicifacies* s.l. has been regarded as a freshwater obligate vector that mainly undergoes preimaginal development in water habitats along the irrigation and riverine systems of Sri Lanka. Sibling species E seems to be robust and is able to undergo preimaginal development in different water habitats that include sand pools and rock pools along river margins, quarries with turbid water, and open and bound wells that vary in limnological characteristics (Table 1) [20]. This adaptation shows that the physical quality of water does not play a role in limiting the preimaginal development of sibling species. A recent study shows that species E can undergo preimaginal development in brackish waters indicating its adaptive adaptation to exploit different habitats and invade new habitats [27]. A possibility of inter-species competition or niche segregation was postulated as none of the collected larval samples were found with both sibling species, although both are sympatric [20].

Limited studies show that species E can live long enough to support the extrinsic incubation period of malaria parasites with an ability to survive longer enough to undergo more than three gonotropic cycles, while species B on average supports less than three gonotropic cycles; although, there was no statistical significance to show that species E has higher fecundity than that of species B (Table 1) [20]. Reasons for differential longevity and fecundity of the members of the Culicifacies complex are unknown but the greater longevity of species E has an implication in malaria transmission since species E fed on *Plasmodium vivax* and *P. falciparum* infected blood was found to have oocysts. None of the species B mosquitoes screened so far were detected with malaria parasites [28]. Although bio-ecological traits have been established for sibling species B and E, no literature is available to describe their impact on malaria transmission in the country.

**Table 1 ijerph-21-00814-t001:** Bio-ecological characteristics of species B and E of the *Anopheles culicifacies* complex in Sri Lanka.

Characteristic	*Anopheles culicifacies*	
	Sibling Species B	Sibling Species E
Mitotic Y chromosome [23]	Acrocentric	Submetacentric
Prevalence [29]	Sympatric	Sympatric
Preimaginal development sites [2,27]	Rock pools, sand pools, quarries	Rock pools, sand pools, wells, irrigation channels, brackish waters
Resting and feeding [29]	Indoor and Outdoor	Indoor and Outdoor
Vector potentiality [29]	Poor vector or non-vector	Vector of *Plasmodium vivax* and *P. falciparum*
Longevity [29]	Tendency to survive less than 3 ovipositions	Tendency to survive more than 3 ovipositions
Insecticide resistance [29]	Resistant to DDT	Resistant to DDT
	Less resistant to malathion	More resistant to malathion
	Susceptible to λ-cyhalothrin and deltamethrin	Susceptible to λ-cyhalothrin and deltamethrin

Modified from [20] with permission.

### 5.2. Anopheles subpictus Complex

*Anopheles subpictus* s.l. is considered to be an important malaria vector due to its role in malaria transmission in many parts of the country including the northern Jaffna district [2,30]. The taxon *An. subpictus* is reported to exist as a species complex comprising four morphologically distinct members, viz. A, B, C and D. These members from India are distinguished by polytene chromosome banding patterns and stage-specific morphometric characteristics [19]. Initial studies based on polytene chromosome characterization corresponding to the inversion pattern of the X-arm of Indian *An. subpictus* revealed the presence of sibling species A and B in Sri Lanka [29]. Later, morphometric characterization compatible with that of Indian samples revealed the presence of all four (A–D) sibling species in Sri Lanka [30,31]. While species B is considered to be a coastal species, species A, C and D are considered to be inland species [32]. Species B is reported to be more salt tolerant than the other inland species [33]. Various morphological attributes in larval and adult forms have been reported for Sri Lankan sibling species [32].

#### 5.2.1. Bio-Ecology of the Sibling Species of Subpictus Complex in Sri Lanka

Differential bioecological traits of morphologically identified members of the Subpictus complex are provided in Table 2. The widespread prevalence of sibling species identified based on morphological characteristics is shown in Figure 1 [30,31,33]. The inland species C peaks in its abundance in November and January, while the coastal species B is abundant in January, December, April and July (Table 2) [30]. While all sibling species are sympatric they show distinct variations in feeding and resting preferences. While species A and C prefer to feed and rest indoors, species B prefers outdoors [31,32]. However, the feeding preferences (anthropophagic/zoophagic) of the sibling species are yet to be clearly established (Table 2). In a limited study from the north-western province in the dry zone, a single specimen of species C was detected with malaria sporozoits [34]. Therefore, the relative role of different sibling species of the Subpictus Complex in transmitting malaria has not been well established.

#### 5.2.2. Recent Development in the Molecular Characterisation of *An. subpictus* Sibling Species and a Need for Taxonomic Reassessment

A phylogenetic analysis based on the DNA sequences of the D3 domain of 28 S ribosomal DNA (rDNA) and the internal transcribed spacer-2 (ITS-2) of mosquitoes morphologically identified as *An. subpictus* sibling species A, B, C and D, resulted in two clades: one clade with mosquitoes identified as *An. subpictus* species A, C, D and the other clade with a majority identified as species B with D3 sequences that were identical to *Anopheles sundaicus* cytotype D. Analysis of ITS-2 sequences confirmed a close relationship between a majority of mosquitoes identified as *An. subpictus* B with members of the *An. sundaicus* complex [35]. Both ITS2 and COI sequences revealed two divergent clades indicating that the Subpictus complex in Sri Lanka is composed of two genetically distinct species, instead of four morphologically distinguished members, that correspond to species A and species B from India. Phylogenetic analysis showed that species A and species B do not form a monophyletic clade but instead share genetic similarities with Indian species A and *Anopheles sundaicus* s.l., respectively [35]. Based on the molecular characterization it has been proposed that the *An. subpictus* complex is also composed of two molecular forms, namely A and B, and the molecular form B is a member of the *An. sundaicus* complex [35,36]. Similar results were reported from the Subpictus Complex of India, which is composed of two molecular forms, namely A and B [37]. Considering the complexity of the morphology-based identification of sibling species, an allele-specific PCR-based identification method has been developed for the reliable identification of molecular forms A and B/*An. sundaicus* s.l. in Sri Lanka [36]. However, considering the role of malaria transmission, a taxonomic reassessment is warranted for samples identified as species B in the Indian sub-continent.

## 6. Other Minor Species Complexes

Apart from the two major malaria vector species, minor vectors such as *An. annularis*, *An. barbirostris* and *An. maculates* also exist as species complexes in the Indian subcontinent [19]. Therefore, it is important to identify the members and examine their role in malaria transmission in Sri Lanka.

*Anopheles annularis* is composed of two sibling species namely A and B in India and species A has been incriminated as a malaria vector [19]. In Sri Lanka, *An. annularis* undergoes preimaginal development in irrigation canals. The recent molecular characterisation based on D3 and ITS-2 rDNA sequence analysis revealed that Sri Lanka *An. annularis* is closely related to Indian species A [38]. This supports the previous report that *An. annularis* functioned as the most efficient vector in north-central Sri Lanka during high malaria transmission periods [15].

The taxon *An. barbirostris* is composed of three sibling species viz., A, B and C in Southeast Asia [19]. Samples collected from the dry zone of Sri Lanka were found to have sporozoites [2]. The molecular characterisation of Sri Lankan samples based on ITS-2 and COI sequences revealed that the Sri Lanka samples are of a new molecular type closely related to *An. barbirostris* s.s. [39].

Even though the presence of *An. maculatus* in Sri Lanka is reported, its sibling species status or role in malaria transmission is yet to be established

## 7. Insecticide Resistance among the Malaria Vectors

### 7.1. Development of Insecticide Resistance

Control of malaria vectors in Sri Lanka has been primarily through the use of synthetic insecticides. Conventional insecticides of three classes i.e., organochlorines, organophosphates and pyrethroids, have been heavily used to date although the fourth class, carbamates, have not been used much in malaria vector control programmes. Organochlorine DDT was introduced as the first insecticide in the 1940s to malaria vector control programmes mainly for indoor residual house spraying (IRS). In 1952, 5 years after the introduction of country-wide DDT spraying, a ten-fold reduction in cases was observed [www.malariacampaign.gov.lk/en/our-services/vector-control (accessed on 23 April 2024)]. Incidence of malaria declined sharply within a decade of its introduction and DDT was heavily used in both the health and agricultural sectors to control insect pests. In 1969, possible resistance to DDT was detected in *An. culicifacies* and *An. subpictus* by the Anti-Malaria Campaign and it is believed that the resistance to DDT contributed in a big way to the extensive island-wide epidemic of malaria in 1967/68. By 1974, DDT resistance was evident in all malaria vector species [www.malariacampaign.gov.lk/en/our-services/vector-control (accessed on 23 April 2024)] [40]. Due to the resurgence of malaria cases, the island-wide development of vector resistance to DDT and environmental concerns, the use of DDT was banned in Sri Lanka in the 1975–1977 period and it was replaced with the organophosphate malathion. In the beginning, the use of organophosphates was restricted to the health sector and carbamates were restricted to insect pest control in the agricultural sector. However, organophosphates were also used subsequently in agriculture. Malathion, which replaced DDT, has been the major insecticide used in adult mosquito control programmes until the introduction of pyrethroids. The main larvicidal insecticide used was the organophosphate temephos.

The development of vector resistance to malathion, and the increased transmission of malaria, led to the introduction of pyrethroids for vector control programmes in Sri Lanka in 1994. The organophosphates malathion, fenitrothion and temephos, and pyrethroids such as λ-cyhalothrin, cyfluthrin, deltamethrin and the pseudo-pyrethroid etofenprox were the major insecticides used for IRS since then. Rotation of chemically unrelated compounds once in every 3–5 years was the strategy adopted by the Sri Lankan malaria vector control programmes to delay the development of resistance. Permethrin (a pyrethroid) was the only insecticide used for the impregnation of mosquito nets. Long-lasting insecticide-treated mosquito nets were also distributed in some provinces [41]. Pyrethroids were also introduced in the agricultural sector, parallel to the health sector.

When malathion was first introduced in Sri Lanka, a 20 min exposure to 5% malathion gave 100% mortality to *An. culicifacies*. The first survivors for this dosage were detected after 2–4 years of malathion spraying in 1979. Resistance to 5% malathion for one hour (standard WHO discriminating dosage) was first observed in 1982 [40]. A study carried out in the late 1990s in a rural area of the Matale district of Sri Lanka showed that *An. culicifacies* had developed 70% and *An. subpictus* 15% resistance to malathion and 26% and 88% respectively for chlorpyrifos. The same study revealed the emerging pyrethroid resistance within a few years of their introduction. For permethrin, cypermethrin and deltamethrin, the two major vectors of malaria showed 26–27%, 0–18% and 0–12% resistance respectively [42]. Another comprehensive study covering five districts in Sri Lanka detailed the status of organophosphate resistance three decades after its introduction and pyrethroid resistance after a decade of its introduction [43]. *Anopheles culicifacies* had developed 0–70% and 0–60% resistance to the organophosphate malathion and fenitrothion, respectively, whereas the respective figures for *An. subpictus* were 51–77% and 38–60%. Resistance levels shown by *An. culicifacies* to pyrethroids were 13–77% to permethrin, 0–93% to deltamethrin and 5–15% to etofenprox. *Anopheles subpictus* showed resistance levels of 33–75%, 8–16% and 0–77% respectively for these pyrethroids. Both species were susceptible to cypermethrin and cyfluthrin, and only *An. subpictus* had 0–42% resistance to lambda-cyhalothrin. All the tested populations of *An. culicifacies* were susceptible to the carbamate propoxur whereas *An. subpictus* showed a resistance of 7–15% to the carbamate showing that the impact of insecticides used in agriculture is minimal for the resistance development in the major malaria vector *An. culicifacies*. Perera et al. [42] also reported that out of eight possible anopheline vectors (i.e., *An. annularis*, *An. barbirostris*, *An. jamesii*, *An. nigerrimus*, *An. peditaeniatus*, *An. tessellatus*, *An. vagus* and *An. varuna*) tested, only *An. nigerrimus* and *An. peditaeniatus* had developed resistance to these insecticides (i.e., 12–43% resistance to organophosphates, 0–71% to pyrethroids and 36–58% to the carbamate propoxur) indicating the probable higher exposure of these two species to the insecticides used in agriculture [44].

The most striking result was for DDT resistance. Three decades after the cessation of DDT usage on the island, both malaria vector species had 38–96% resistance to DDT and other anopheline species had 43–93% DDT resistance [43]. Preservation of high resistance to DDT has been detected not only in malaria vectors but also in other Sri Lankan insect pest populations, such as the populations of dengue vectors *Aedes aegypti* and *Ae. albopictus* [45], Japanese encephalitis vectors *Culex tritaeniorhynchus* and *Cx gelidus* [46], bed bug *Cimex hemipterus* [47], sand fly *Phlebotomus argentipes* [48], cattle tick *Rhipicephalus microplus* [49] and a range of agricultural insect pests [50]. It has been suggested that underlying mechanisms, especially the enhanced GST-mediated metabolism, initially selected by exposure to DDT, have been subsequently selected by exposure to organophosphates and pyrethroids [42,51,52].

### 7.2. Mechanisms of Insecticide Resistance

Mechanisms underlying insecticide resistance are mainly the enhanced metabolism and insensitive target sites in the insect body. Enhanced insecticide metabolism is brought about by the increased activity of three classes of metabolic enzymes, namely esterases, glutathione S-transferases and monooxigenases. Increased activity is a result of either quantitative or qualitative changes in the enzyme. Target site insensitivity is due to gene mutations which make target proteins incapable of binding to insecticides but capable of performing their normal physiological functions within the insect body [53,54].

Voltage-gated sodium channel proteins are the target site of both the pyrethroids and the organochlorine DDT. Its insensitivity to insecticides (known as *kdr* type resistance) due to an L1014F mutation has been reported by sequencing a fragment of the gene from etofenprox-resistant *An. subpictus* from Anuradhapura, Sri Lanka [55]. However, it has been reported that the DDT resistance of malaria vectors initially developed due to a GST-based resistance mechanism [12,13]. Resistance to malathion among the populations of both malaria vectors was initially due to malathion carboxylesterases, which are qualitatively different esterases with higher catalytic centre activity towards malathion [40,42,56]. It was later revealed that the mechanism with increased quantities of general esterases/carboxylesterases, a mechanism different from that of malathion carboxylesterases but commonly found among *Culex* mosquitoes, has also been developed in Sri Lankan malaria vectors. These esterases are capable of metabolising organophosphates rich with ester bonds at a faster rate [54]. By 2004, major malaria vectors and other possible vectors of malaria had developed metabolic resistance with increased activity of esterases, GSTs and monooxygenases, and altered acetylcholinesterases (acetylcholinesterase is the target site of organophosphates and carbamates) and kdr type resistance [43,55]. Involvement of monooxygenases and esterases in pyrethroid and organophosphate resistance in these anopheline species has also been shown by synergistic bioassays using the monooxygenase inhibitor piperonyl butoxide (PB) and the esterase inhibitor triphenyl phosphate (TPP) respectively [41,43].

Several studies have demonstrated the heterogeneity of malaria vector resistance to insecticides [42,43]. This may be due to the presence of sibling species with different insecticide cross-resistance spectra. *Anopheles subpictus* species A (morphologically identified) is abundant predominantly in inland regions and moderately resistant to organophosphates whereas species morphologically identified as B are more confined to the coast and resistant to permethrin in Sri Lanka [29,57]. Both sibling species B and E of *An. culicifacies* were found fully susceptible to lambda-cyhalothrin and deltamethrin, but resistant to DDT and partially resistant to malathion; although, the resistance to malathion was higher in species B [27]. Studies on insecticide resistance on isolated sibling species have only been attempted with the members of the *An. subpictus* species complex. All sibling species were highly resistant to DDT. Species B was more susceptible to all the tested insecticides than the sibling species A, C and D. The authors have attributed this difference to the lesser exposure of species B to the indoor residual spraying of insecticides. Species A, C and D had developed metabolic resistance and insensitive acetylcholinesterases to varying degrees. However, increased monooxygenases were present only in the subspecies A [58]. Susceptibility of *An. subpictus* species B (regarded as a member of *An. sundaicus* complex) to insecticides has been confirmed recently as well [59].

Intensive IRS coverage in foci and the distribution of long-lasting insecticide-impregnated bed nets (LLIN/ITN), and other complementary measures contributed to Sri Lanka’s march towards a malaria-free island in 2013. However, the re-introduction of malaria transmission by infected overseas travellers is possible due to the prevalence of potent malaria vectors. Preventive measures are taken by intense surveying of vector abundance around imported malaria cases. Introduction of ITNs and IRS in risk areas are still in operation [www.malariacampaign.gov.lk/en/our-services/vector-control (accessed on 23 April 2024)]. Knowledge of the insecticide resistance status among anopheline vectors is important if large-scale vector control happens to be reintroduced on the island.

## 8. Recent Invasion of *Anopheles stephensi*, Major Asian Malaria Vector, to Northern Sri Lanka

The recent spread of *Anopheles stephensi*, the major Asian malaria vector, to African countries and Sri Lanka is of great concern to control malaria [60]. The presence of *An. stephensi* in Sri Lanka was first reported on northern Mannar Island in 2017 [10]. Later its spread was detected in the northern Jaffna city [61]. The biotype identification of invaded samples revealed the presence of a potential urban malaria vector type form developing in rural domestic wells, domestic wells and overhead tanks in urban areas. The DNA sequence analysis based on two mitochondrial (COI and cytb) and one nuclear (ITS-2) markers showed that *An. stephensi* collected from Mannar Island are genetically identical and related to *An. stephensi* populations in the Middle East and Indian subcontinent [61].

*Anopheles stephensi* of Jaffna city was found to be resistant to DDT, malathion and deltamethrin and was able to undergo preimaginal development in brackish water of 3–5 g/L salt, polluted water and alkaline water [62]. These bioecological traits indicate that while the species can expand its range along the coastal line, the controlling efforts would be challenging.

After the detection of *An. stephensi* in the southern coastal areas of south India in 2001, it was postulated that *An. stephensi* may spread across the narrow Palk Strait to Sri Lanka [63]. The spread of *An. stephensi* into African countries and Sri Lanka, among urban water habitats such as wells and cemented tanks, may provide an example of the postulated anthropogenically-induced adaptation to invade new territories in mosquito vectors [64].

The absence of indigenous malaria since 2012 makes the situation impossible to associate the role of invaded *An. stephensi* population with local malaria transmission. However, continuous monitoring of the presence and invasion of other parts of the country is warranted.

## 9. Conclusions

Sri Lanka (then Ceylon) was on the verge of eliminating malaria in 1963 when only seventeen cases were reported of which six were indigenous. Due to negligence, the country missed an opportunity and it took nearly five decades of hard work to eliminate malaria thereafter.

Now, Sri Lanka is in the prevention of the reintroduction phase which is implemented by the National Anti-malaria Campaign with a defined policy and practice to prevent malaria reintroduction. The implementation of practices such as monitoring imported malaria cases and the prevalence of malaria vectors are still challenging due to public movement to malaria-endemic countries, the presence of vectors and their adaptation to changing environments, the invasion of potential malaria vectors, and the continuous reports of imported malaria cases. While Sri Lanka stands out to be an exemplary nation that eliminated malaria mainly due to coordinated public health activities, challenges are still ahead to retain its malaria-free status. Continuous studies to update the knowledge on vectorial capacity of anopheline sibling species, genetic structure and their differential insecticide resistance status are essential to face these future challenges.

## Figures and Tables

**Figure 1 ijerph-21-00814-f001:**
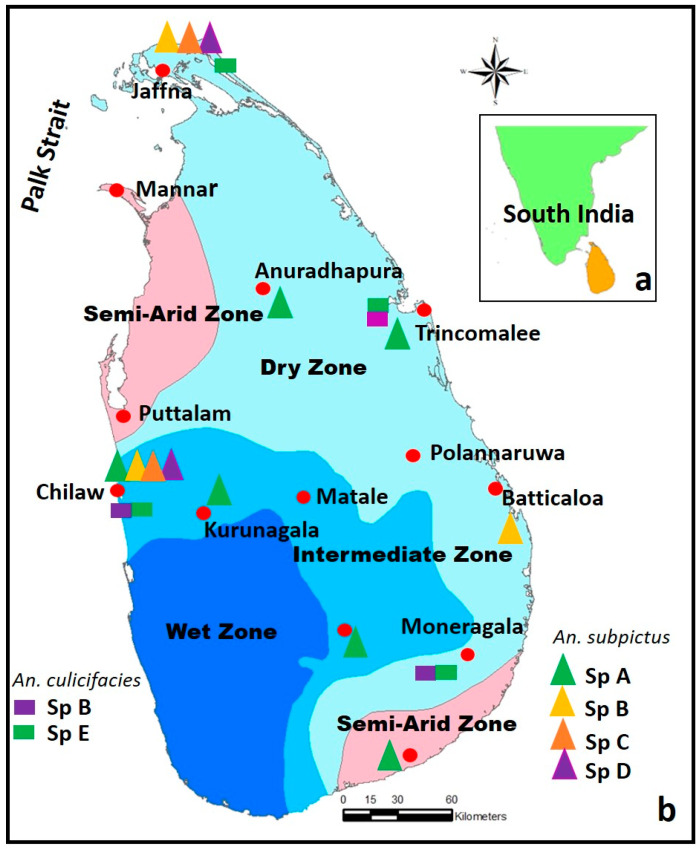
The map shows (**a**) the location of Sri Lanka in relation to south India; (**b**) the climatic zones of Sri Lanka and the distribution of sibling species B and E in the *Anopheles culicifacies* complex and species A, B, C and D in the *Anopheles subpictus* complex.

**Table 2 ijerph-21-00814-t002:** Differential bioecological traits of morphologically identified sibling species of *Anopheles subpictus* complex.

Characteristics	*Anopheles subpictus*			
	Species A	Species B	Species C	Species D
Prevalence	Sympatric	Sympatric	Sympatric	Sympatric
Preimaginal development	Inland fresh water	Coastal fresh water and brackish water	Inland fresh water	Inland fresh water
Resting and feeding	Indoor	Outdoor	Indoor	Indoor and outdoor
Peak abundance	Not known	January, April, December and July	November and January	Not known
Vector potential	Not know	suspected	Vector	Not known
Insecticide resistance	Resistant to DDT and malathion, and highly resistant to pyrethroids with L1014F *kdr* mutation in Jaffna population	Resistant to DDT, less resistant to malathion and susceptible to pyrethroids	Resistant to DDT and malathion, and highly resistant to pyrethroids	Resistant to DDT and malathion, less resistant to pyrethroids

Modified from [20] with permission.

## Data Availability

The datasets presented in this article are from published articles which are appropriately quoted in the text.

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
