# Peer review of "Taxonomy, Bio-Ecology and Insecticide Resistance of Anopheline Vectors of Malaria in Sri Lanka"

_ijerph, 2024, doi:10.3390/ijerph21070814_

Round 1

Reviewer 1 Report

Comments and Suggestions for Authors

I really liked the review article containing important about the taxonomy of anophelines that occur in the different climatic regions of Sri Lanka, with emphasison molecular taxonomy that separates the species comprising the anopheline complexes containing primary and secondary malaria vectors for the study region. 

Articles wre also reported that addressed the use of insecticides in controlling anopheline populations and the resistance of these mosquitoes over the period of use of different categories of insecticides.  Bio-ecological aspects were also coverred, always with an emphasis on the Anopheles culicifacies complex and the Anopheles subpictus complex. It also reports the recent 2017 invasion of Anopheles stephensi in a northern city of Sri Lanka, whose resistance was recorded for all three classes of insecticides used in malaria control. 

It is a good review work, very useful for peers working in research lines related to the topic.

What was not mentioned is that the goal was to write a review article.

The authors conducted an extensive and updated literature review. However, they did not mention the selection criteria for the references used in the writing of the article. Meaning, whether therewere inclusion and exclusion criteria for bibliographic references. 

The authors must correct the name of the species Anopheles culicifacies, in line 143, which was writtem incorrectly. The authors shoud also correct the species name An. stephensi in line 421.

The conclusion section should be from the authors of the article and there shouldn't be citations, such as those in numbers 69 and 70. Such references can be cited in the body of this article except in the conclusion.

I believe there are missing citations for the texts contained in lines 24 to 28, 35 to 38, and 406 to 413.

Those were the observations I had for the moment.

Author Response

We thank the reviewer 1 for making constructive comments on the MS. Herewith I am submitting our responses to the comments made by the Reviewer 1

Reviewer 2 Report

Comments and Suggestions for Authors

Sri Lanka, once endemic for malaria, reported its last indigenous case in 2012 and was declared malaria-free in 2016. The primary and secondary vectors of malaria, Anopheles culicifacies and An. subpictus, exist as species complexes with varying bio-ecological traits and insecticide resistance. Despite the cessation of DDT use decades ago, resistance persists in malaria vectors and other insect pests. This article provides a comprehensive update on the taxonomy, bio-ecology, and insecticide resistance of these vectors, highlighting the challenges in maintaining the malaria-free status and the recent invasion of An. stephensi, a major concern for malaria control. Here I have a few comments for the authors to address.

1) the authors should add supporting data or citations in several locations. For example,

Line 106 to 113:

The discussion on the Anopheles culicifacies complex mentions genetic variations but lacks specific studies or data to support these genetic differences and their implications.

Line 136 to 153:

The bio-ecological traits of species B and E are described, but there is no reference to studies or data that document these traits and their impact on malaria transmission.

Line 418 to 439:

The spread of Anopheles stephensi is described, but the text lacks detailed evidence or studies showing how this spread has impacted malaria incidence in the affected regions.

2) The authors should better organize the structure of the manuscript.

- Transitions Between Sections:

The transitions between sections are abrupt, with little to no connective text to guide readers from one topic to the next. This abruptness can make the review feel fragmented and disjointed. For instance, the author should consider a more natural transition between Section 1, "Malaria in Sri Lanka," and Section 2, "Anopheline Species in Sri Lanka." Similarly, smoother transitions should be implemented between other sections to enhance the overall flow and coherence of the review.

- Mixed Content:

Information on historical context, current status, and technical details about malaria vectors and insecticide resistance is intermingled, which disrupts the logical flow. For example, discussions about historical malaria epidemics and recent trends are not clearly separated from technical descriptions of mosquito species and resistance mechanisms.

- Poor Integration of Figures and Tables:

Figures and tables are not well-integrated into the text. They are mentioned without proper introduction or discussion, which makes it difficult for readers to understand their relevance. For instance, Figure 1 is referenced without adequate explanation of what it represents or how it contributes to the discussion.

3) The conclusion should be strengthened to include a succinct summary of the main findings of the review. Additionally, it should highlight the implications of these findings for future research, policy, and practice. Finally, it should offer recommendations and measures to control malaria, particularly focusing on efficient strategies for future elimination.

Minor:

Line 6: "Deapartment" should be "Department"

Line 8: "A Sri Lanka" should be "Sri Lanka"

Line 14: "highlight the challengers" should be "highlights the challenges"

Line 16: Missing space in "5;sibling" should be "5; sibling"

Line 17: Missing space in "8;Sri" should be "8; Sri"

Line 20: “65,525 km2” should be “65,525 km2

Line 53: "culicifcaies" should be "culicifacies"

Line 116: "detected not only" should be "detected not only"

Line 139: "indoor" should be "indoors"

Line 142: "culicifcaies" should be "culicifacies"

Line 145: "preimaginal development" should be "pre-imaginal development"

Line 149: "adaptive radiation" should be "adaptive adaptation"

Line 151: "samples found with both sibling species" should be "samples found with both sibling species"

Line 293: "with in a decade" should be "within a decade"

Line 374: "higher catalytic centre activity" should be "higher catalytic center activity"

Line 376: "mechanism different from that of malathion" should be "mechanism different from that of malathion carboxylesterases"

Comments on the Quality of English Language

A modest revision of the language is required.

Author Response

We thank the Reviewer 2 for making constructive comments on the MS. Herewith I submit our responses to the comments made by the Reviewer 2.

Round 2

Reviewer 2 Report

Comments and Suggestions for Authors

The authors have addressed most of the concerns and this manuscript is ready to go.